# Human–Deer Relations during Late Prehistory: The Zooarchaeological Data from Central and Southern Portugal in Perspective

**DOI:** 10.3390/ani14101424

**Published:** 2024-05-10

**Authors:** Nelson J. Almeida, Catarina Guinot, Inês Ribeiro, João Barreira, Ana Catarina Basílio

**Affiliations:** 1CHAIA—Centre for Art History and Artistic Research, IN2PAST—Associated Laboratory for Research and Innovation in Heritage, Arts, Sustainability and Territory, Department of History, Colégio do Espírito Santo, University of Évora, 7000-645 Évora, Portugal; i.isabel.rosaribeiro@gmail.com; 2Uniarq—Centre for Archaeology, School of Arts and Humanities, University of Lisbon, 1649-004 Lisbon, Portugal; 3Department of History, Colégio do Espírito Santo, University of Évora, 7000-645 Évora, Portugal; csrguinot@gmail.com; 4Museu Rainha Dona Leonor, 7800-131 Beja, Portugal; joaombarreira@gmail.com; 5O Legado da Terra, Cooperativa de Responsabilidade Limitada, 7800-651 Nossa Senhora das Neves, Portugal; 6CEAACP—Centro de Estudos de Arqueologia, Artes e Ciências do Património, 3000-607 Coimbra, Portugal; 7ICArEHB—Interdisciplinary Center for Archaeology and the Evolution of Human Behaviour, University of Algarve, 8005-139 Faro, Portugal; catarinasbasilio@gmail.com

**Keywords:** red deer, Late Prehistory, southern Portugal, human–animal relations, osteology, zooarchaeology

## Abstract

**Simple Summary:**

Human–animal relations during Prehistory are an increasingly discussed issue among scholars. Cervids are a symbolically prominent species among hunter–gatherers but are also of relevance to pastoralists. We present a synthesis of available data regarding human–deer relations in Portuguese Late Prehistory. This is achieved by examining deer abundance in subsistence, together with their inclusion in social practices such as food-sharing and their involvement in structured depositions in both funerary and non-funerary contexts. Additionally, the study investigates the utilization of deer remains for the creation of artifacts and their depiction in pottery decoration, figures, and rock art. The dynamics of deer prevalence in subsistence have varied diachronically and synchronically in the periods and areas under study. Although the reasons and meanings for this behavior continue to be discussed, their sociocultural and ideological relevance seems to have persisted in early agropastoral and metallurgical societies, even as they undergo change.

**Abstract:**

Human–animal relations have been a fruitful research topic worldwide. The importance of deer in hunter–gatherer societies is undeniable, with cervids being commonly found in archaeological and past artistic records, with a notable amount of information recovered in the Iberian Peninsula. This relevance continues during Late Prehistory, but the attempt to discuss cervids under broader perspectives and based on different types of data is not as common. We intend to approach human–deer relations in Central and Southern Portuguese Late Prehistory by considering the zooarchaeological records, both deer abundance in faunal spectra and their presence in “meaningful” assemblages and structured depositions, as well as the use of deer and deer body parts in other socio–cultural and ideological practices. The synthesis of available data shows that human–deer relations changed through time and space, with different abundances related to hunting depending on chronology and geography. The use of deer or their body parts as a resource of symbolic nature also varied, being included in food-sharing events, offerings, structured depositions, and graphic representations. Changeability is part of the different relationships, ontologies, and cosmogonies that humans and deer developed in the Late Prehistoric relational world.

## 1. Introduction

The study of human–nonhuman relations during Prehistory is a complex and rather underdeveloped area of research. Recently, several colleagues have dedicated themselves to this topic at different locations and periods. Among the species that have been studied under these perspectives in traditional societies, cervids have gained special interest from researchers through time (e.g., [1,2,3]).

Focusing on the Western Iberian Peninsula, both red deer (*Cervus elaphus*) and roe deer (*Capreolus capreolus*) existed during Late Prehistory. However, roe deer are generally absent or vestigial in zooarchaeology records. The former is more frequent and normally occupies a large variety of habitats, such as scrubland, open deciduous, and mixed woodland, being a highly adaptable species. In contrast, the latter is generally related to mosaic habitats with woodlands near scrubland and meadows. Cervids had an important place in Prehistoric societies, not only concerning their palaeoeconomical input but also their sociocultural and ideological significance, as can be seen in the material culture and their depictions in rock art [4,5] or other expressions present, for example, in pottery decoration. The study of human–nonhuman relations gains an added interest when major changes occur in societies, with these studies being involved in larger uncertainty, as is the case of the transition to farming. The neolithization processes mark several changes in the economic, technological, social, cultural, and ideological spheres of human societies. Ingold [6] argues this would have resulted in profound changes in human–animal relations toward dominance, contrary to the previous trust relationship that existed in hunter–gatherer groups. Even considering that pure utilitarian modern taxonomies and categorizations have problems when applied to a more diverse relational world if dealing with cervids, we are focusing on animals that were not domesticated. Thus, previous forms of relation could be maintained [7,8], emphasized, or even reinvented and adapted to the narratives being constructed. This adds to the complexity of possible relationships, ontologies, and cosmogonies that humans could have had with or involving cervids, their image or parts of these during Late Prehistory.

Colleagues have already highlighted the importance of red deer in the rock art of Holocene hunter–gatherers and Late Prehistory pastoralists from the Southwestern Iberian Peninsula (see [5,9]). Although the study of human–animal relations in the region can be seen as somewhat underdeveloped, interesting discussions have arisen looking at specific species or wider topics (e.g., [7,10,11,12,13,14], with cervids being mostly discussed concerning rock art (but see [15])).

In this paper, we present an appraisal of available zooarchaeological data for Central and Southern Portugal’s Late Prehistory, i.e., Neolithic and Chalcolithic (circa mid-6th to the end of the 3rd millennium BCE). We intend to discuss human–deer relations within this geographical and chronological framework, attempting to see if changes have occurred and relate them to wider data and problematics. Therefore, zooarchaeological information is the starting point, but comparisons with material culture and rock art are also discussed to present a more informed perspective on human–deer relations.

## 2. Materials and Methods

Extensive bibliographical research was conducted on available zooarchaeological data for Central and Southern Portugal Late Prehistory. Non-funerary sites/contexts with macrofauna (*Equus*, *Bos*, *Sus*, *Cervus*/*Capreolus*, *Ovis*/*Capra*) were considered if the number of identified specimens for these species was >50. This number was selected considering the small number of the majority of Early and Middle Neolithic published faunal assemblages, but still with an amount of information that can be viable for discussion. When larger assemblages are available, the discussion will focus on them. Although red deer and roe deer, respectively, *Cervus elaphus* and *Capreolus capreolus*, are considered together, when present, the roe deer is vestigial.

Funerary contexts or instances of structured deposition of deer remains, regardless of the number of remains, are mentioned in the results section. To complement the analysis, research on the use of deer bone and teeth for other purposes (e.g., artifacts, pendants) was conducted. The results obtained are presented by data type and then discussed together with other pieces of evidence at both regional and supra-regional scales.

This resulted in a total of 31 archaeological sites, mostly located in the Estremadura and Alentejo regions of Portugal (Figure 1). These sites are divided into 43 different chronological phases, comprising the Early Neolithic (circa 5500–4500 BCE, *n* = 7), the Middle Neolithic (circa 4500–3500/3200 BCE, *n* = 6), the Late Neolithic (circa 3500/3200–3000 BCE, *n* = 9), and the Chalcolithic (circa 3000–2200 BCE, *n* = 21) (Table 1). Relative chronologies based on archaeological information are occasionally followed when absolute dates are not available. Discussed sites and contexts have different chronological reliabilities but these are not expected to decisively impact the topics under discussion. The use or lack of sediment sieving is not expected to decisively impact the results presented. Even if information on this is not always given, Early and Middle Neolithic sites are more commonly sieved. In contrast, Late Neolithic/Chalcolithic sites probably include several contexts where sieving was not implemented. The archaeological contexts are described when needed to better characterize the depositional environments and artifactual associations. Regarding the discrepancy in the number of sites per chronology and region, this is not the appropriate place to explore this matter further. However, it is worth noting that the archaeological visibility and the historiography of research have biased the available information.

## 3. Results

### 3.1. Deer Abundance in Faunal Spectra

Available Early Neolithic data comes almost exclusively from sites in Estremadura (5 out of 6). These show how cervids are completely absent or vestigial on some open-air sites, namely Lameiras (Early Neolithic = 2%; Evolved Early Neolithic = 0%) and Carrascal (0%) (Figure 2). Encosta de Sant’Ana, also located in the Lisbon Peninsula, has a slightly higher prevalence of deer remains (8%). Sites located in the karstic area of the Limestone Massif have higher values—Caldeirão cave (10%) and Pena d’Água rock-shelter (32%)—as is the case for Vale Boi (24%), the only site included located in the Algarve region, in southernmost Portuguese territory.

The number of sites related to the Middle Neolithic in the study area follows the same patterns seen for the Early Neolithic with fewer sites. A common trend in the majority of them is that they are located in the Estremadura Limestone Massif (5 out of 6). Of these, we count animal remains recovered in caves (*n* = 3), rock shelters (*n* = 1), and open-air occupations (*n* = 1). The latter site, Costa do Pereiro, has the highest prevalence of deer (67%) if excavation artificial levels 1 and 2 are considered together. The Pena d’Água rock shelter (27%) showed important but comparatively lower values of deer.

The collections recovered in cave necropolises but considered as showing a non-funerary use of these sites have divergent scenarios. On the one hand, Nª Sª das Lapas and Cadaval (layer C) caves have intermediate values for deer remains, 11% and 18%, respectively. Still not completely published, deer is absent from the assemblage from Casais da Mureta, which is dominated by caprine, similar to the remaining sites mentioned in this area except for Costa do Pereiro, where they correspond to the second most common group, behind red deer. The sole site published outside of Estremadura is the ditched enclosure of Perdigões in inland Alentejo. This assemblage shows that deer (28%) were still relevant during the Late Middle Neolithic, with values comparable to Pena d’Água rock shelter.

Late Neolithic is a period during which larger sites are known in the area under study, impacting the number of published faunal samples and the number of remains of some of these assemblages. While Estremadura continues to be of major importance, with 5 out of 9 sites discussed, the number of sites in the Alentejo (*n* = 4) increased. The former corresponds to open-air settlements, including walled sites, while in the latter, we start to see the significance of ditched enclosures, which will be even more evident later during the Chalcolithic. Focusing on the Late Neolithic collections, we have Lameiras (1%), Belas (6%) and the large sites of Leceia (1%), Penedo do Lexim (0%), and Espargueira/Serra das Éguas (4%), the latter corresponding to a Late Neolithic/Chalcolithic open-air site, but considered mostly from the Late Neolithic. All of these demonstrate that deer hunting was not common. In the case of inland Alentejo, both Juromenha 1 (6%) and Perdigões (9%) have values below 10%. At the same time, we start to see samples where deer are more abundant, especially at the Barranco do Xacafre (18%). Moinho de Valadares (32%) shows high values but is a smaller assemblage.

The Chalcolithic period is where major differences are seen in the main landscape units’ zooarchaeological records [34,54]. Open-air settlements, including walled sites, comprise a total of 8 sites with data for 10 phases. We can mention the smaller samples obtained for Castro de Chibanes (1A1/2—4%) and Castro da Fórnea (17%) or the later ones from Vila Nova de São Pedro (8%) and Castro de Chibanes (IC—3%). Comparatively larger collections were recovered at Ota (9%), Castro do Zambujal (4%), Leceia (1%), Penedo do Lexim (1%), and the later phase from Castro do Zambujal (4%). If one disregards the much smaller sample from Castro da Fórnea, the only exception to a common trend of deer vestigial frequencies is Castro de Columbeira, located in a slightly different landscape than the other sites mentioned.

Alentejo has 7 sites corresponding to 10 Chalcolithic phases with faunal assemblages included in this analysis. The information comes from various sources, including walled enclosures and, most notably, ditched enclosures. However, there is a clear predominance of samples collected in negative features, i.e., pits and ditches. Porto Torrão is a very large ditched enclosure with a recently published Chalcolithic small sample (7%). The older studies concerning Chalcolithic (3%) and Late Chalcolithic (23%) do not provide the number of specimens or absolute data, hindering their proper discussion. However, a tendency towards a higher prevalence of deer is seen in the latter. Other small collections show different abundances, ranging from 4% at the small Santa Vitória ditched enclosure to 21% at the Alto de Brinches 3 pit field, with the sample of the walled settlement of Monte da Tumba having 14%. Larger assemblages recovered from the settlement of Mercador (9%) show differences from São Pedro (I–IV—36%) and the Chalcolithic sample from the ditched enclosure of Perdigões (10%), for which we have assemblages dated to the Chalcolithic–Early Bronze Age transition (30%).

### 3.2. Deer in “Meaningful” Assemblages and Structured Depositions

“Meaningful” assemblages and structured deposition of deer remains have been described for only a few sites. In our opinion, this is related to the difficulty of including these possibilities in the description and interpretation of the archaeological records within Portuguese archaeology discourses. Still, other aspects have influenced this, such as funding limitations. This probably impacted more decisively on older excavations and faunal reports, especially from the Estremadura, for which we have large and important assemblages, mostly obtained at the end of the 20th century and the first decades of the 21st century. Hence, the majority of information available on these topics is found in the Alentejo publications. Still, even here, the constraints of contract-based archaeology must have resulted in an information bias regarding these issues.

At least two food-sharing or feasting events have been suggested, characterized by varying frequencies of deer remains. These events were described in Chalcolithic contexts from ditched enclosures located in the Alentejo. It is currently undeniable, in the face of the growth in published examples, that sociocultural and ideological practices involving, but not limited to, animal remains have occurred in Late Prehistoric funerary and non-funerary contexts. An example is pit 13 from Monte das Cabeceiras 2 [56], especially Stratigraphical Unit (herein SU) 1300, located at the top of the pit. Although it is not clear if remains from other SUs could be part of this event, SU 1300 had 316 faunal remains, mostly undetermined (*n* = 264). Of these, fragments from animals >100 kg are prevalent (*n* = 127), while the identified remains (*n* = 52) show higher values of bovine (*n* = 32) and swine (*n* = 9), but also caprine (*n* = 5), hare (*n* = 1), and red deer (*n* = 5). If we take into account all the SUs from the pit, 8 of the 70 (11%) remain identified as species from red deer (11%) but correspond to a minimum number of individuals of 4 with a clear selection of body portions. Almost at the bottom of this pit, a primary inhumation of a human female adult in an uncommon prone position was registered, and on the bottom SU, a deposition of a deer antler. This pit is linked to pit 54 through a small connection, filled with sediment, small stones, and horizontally deposited schist slabs in two successive rows. Pit 54 had remains from at least one human adult, including scarce anatomical connections. Independently of the type of context (feasting/offerings/depositions), it is clear that deer had a symbolic value that gained relevance when some social event or practice occurred.

This is evidenced by the association between antler depositions and funerary contexts. Still, this context raises other questions since it is not only that deer body parts had a symbolism; the social practice of food-sharing itself is imbued in behaviors that go beyond mere subsistence, and deer remains were part of this. This context and the remains recovered do not have parallels in regional Late Prehistory. Animal anatomical connections were not registered, and long bones had fresh breakage indicators, showing the butchery of the animal remains. This assemblage was related to a feasting event [56]. We argue that deer did not need to have a special symbolism in this case (they could). Still, the social practice itself has symbolism related to the context and unusual number of wild animal body parts, in which deer are included.

A feasting event was proposed for the Chalcolithic/Early Bronze Age transition “cairn 1” and the subjacent pit 79 of Perdigões ditched enclosure [57] (Figure 3). The majority of animal remains were recovered from pit 79 (*n* = 1644) in comparison to the “cairn” (*n* = 80). Among taxonomically identified remains, red deer are prevalent (*n* = 112, 57%; 64% if only macrofauna are considered), corresponding to at least 8 individuals (MNI). Abundant taphonomical indicators of butchering, processing, and consumption were recorded in the assemblage, once more with a prevalence of wild species. The aspects discussed previously also apply to this event, but in this case, deer are the prevalent species. One cannot discuss deer symbolism if these types of special contexts are not considered. In these cases, the symbolism is not necessarily given by the species because the context itself demonstrates the existence of already symbolic social practices.

Concerning structured depositions, their inference in Portuguese archaeology is very biased, depending on research agendas, methodologies, theoretical positions, and other characteristics of the development of archaeological excavations. However, it is worth noting some interesting cases where deer remains were considered structured depositions.

Red deer antler remains at Alto de Brinches 3, including at least two portions of naturally fallen antlers, were considered structured depositions, possibly offerings [46] (Figure 3). The authors do not discuss this further, but the anatomical representation presented regarding red deer is quite striking since they comprise almost exclusively antler fragments (29 out of 30 NISP). They were deposited together with pottery and lithic artifacts, accompanying caprine, bovine, and dog body parts. Among these, at least one was a naturally fallen antler.

In addition to the already mentioned antler from Monte das Cabeceiras 2 pit 13, an uncommon deposition of horns and antlers of cervids, bovines, and caprines (due to frequency but also species diversity—NISP = 14) was found accompanying a more common chalcolithic faunal profile in pit 16 [56] (Figure 3). Cervids, mostly red deer and one roe deer antler—the latter being rarely registered in this region in the Late Prehistory—reached 17% of this collection of taxonomically identified macrofauna (14 out of 81). In addition to two humeri and one loose incisor, all cervid remains were antlers. This is a context where antlers and horns were selected and deposited, something unrecorded in this region and period. However, the context itself reinforces the interpretation that these were not simple caches of raw materials or food waste but instead suggests an intentional deposition. Other examples are recorded, for example, a complete red deer antler that was part of the depositions registered in the second phase from Perdigões Tomb 2 [58] (Figure 3), thus accompanying funerary contexts. Finally, the presence of complete/partial deer skeletons and the predominance of isolated or deer remains in the infillings of elongated sub-rectangular and “bone” shaped negative structures is worth mentioning. Though an interesting debate can be had regarding the functionality of these structures, deer remains were recorded at the Late Prehistoric inland Alentejo sites of Monte das Aldeias, Monte do Outeirinho Novo, and Vale Frio 2 [59].
Figure 3Some examples mentioned in the text are as follows: (**a**) Deposition of red deer antler at SU 1600 from Monte das Cabeceiras [57] (photo by Nelson Borges). (**b**) “Cairn” 1 from Perdigões and (**c**) the pits below the “cairn”, with the emphasis on pit 79 (photos by António Valera, ERA Arqueologia S.A.) [56]. (**d**) SU 235 from Alto de Brinches 3 with the deposition of animal body parts, including red deer antlers (#1), remains of caprine (#2), bovine (#3), dog (#4) and pottery (#5) and (**e**) another antler with the burr portion preserved from SU 7 [46]. (**f**) Reuse phase from Perdigões tomb 2 showing human remains (photo by António Valera, ERA Arqueologia S.A., [58]) and (**g**) the red deer antler deposited [60]. (**h**) Detail of depositions from the upper phase of infilling SU 183 from Perdigões hypogeum 1, containing pottery and a red deer cranial portion with attached antlers [61].
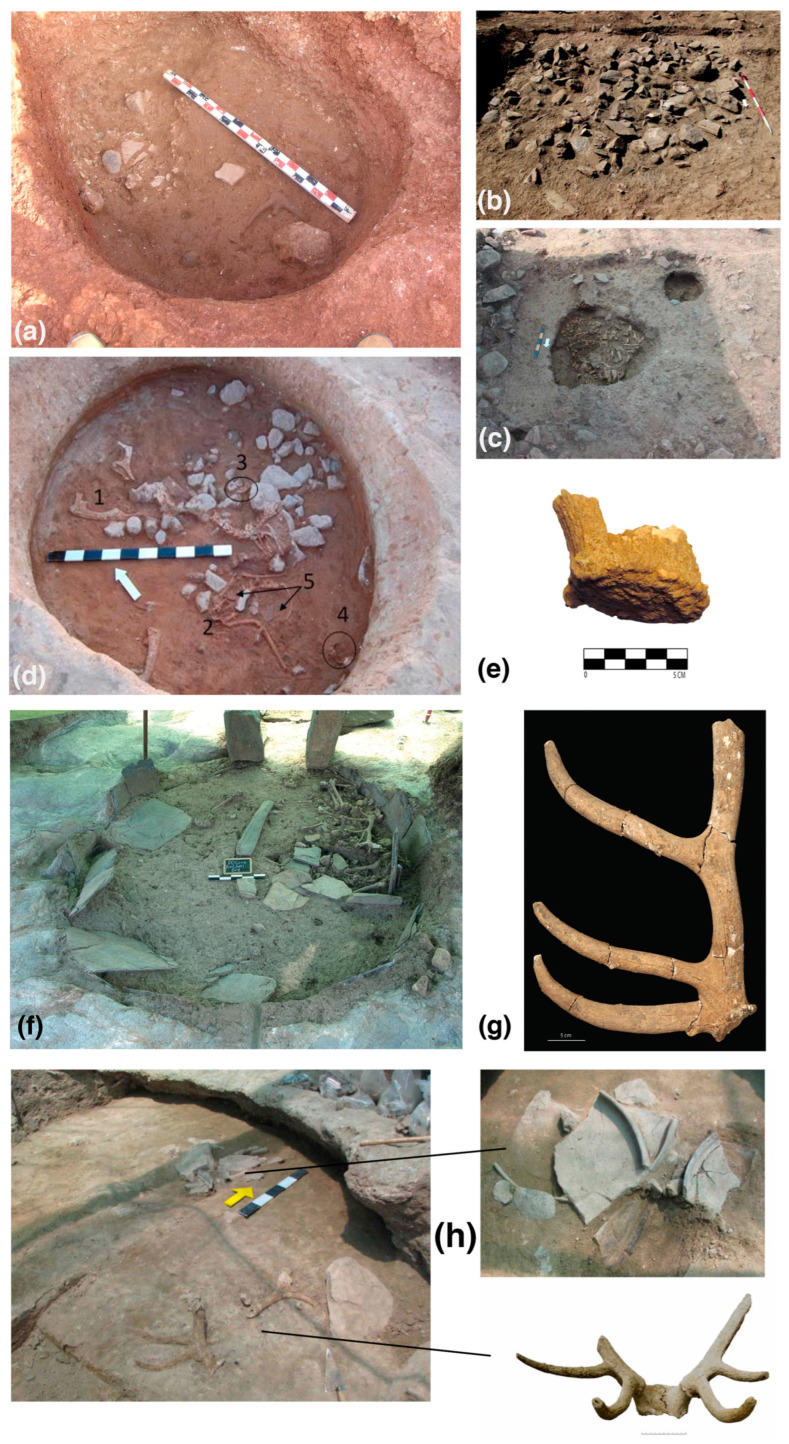


### 3.3. Deer and Other Material Aspects

Perforated red deer canine beads or oval-shaped imitations made with bone, shell, and stone in association with the onset of the Neolithic in Portugal have been found in Nª Sª das Lapas [62] and Almonda [63] (Figure 4) funerary caves in Estremadura, and Padrão 1 [23], and Rocha das Gaivotas [64] in the Algarve.

Regarding the use of deer elements as “technological” material, one must consider preservation issues, but also the lack of publications dealing with bone artifacts (but see [64]). Examples have been mentioned for Neolithic and Chalcolithic sites, but there is a clear tendency towards the use of domesticated animal remains to the detriment of wild animals (e.g., [68,69,70,71]). Even if the use of red deer long bones for the production of artifacts generally made with other species is also registered [72], the use of antlers for these purposes is normally considered sporadic (p. 39, [68]). Nevertheless, awls and modified/worked antlers have been mentioned for Neolithic and Chalcolithic sites, an example being the Bom Santo Middle Neolithic cave necropolis. A polished metapodial with a V-shaped notch and five perforations on the distal portion, considered to have zoomorphic traits and known as the “flute” is so far unparalleled in Portuguese Late Prehistory [24]. Awls, handles, and small “containers” made from red deer long bones and antlers are documented at Leceia [73,74].

Some unparalleled cases in Portuguese prehistory (see p. 39, [68,75]), as far as we know, are the Chalcolithic ivory deer figurine from Perdigões Tomb 2 [65,76] (Figure 4)—where the already mentioned red deer antler deposition was registered—or the cranial portion with antlers attached from an adult red deer, interpreted as a mask or accessory, recovered at the Late Neolithic hypogeum 1 from the same ditched enclosure [61] (Figure 3). The decorated plaques/loom weights from Vila Nova de São Pedro, specifically those decorated with schematic deer motifs, are also worth mentioning [66] (Figure 4). There is an ongoing discussion regarding whether these are related to weaving or other types of practices/use.

Phalange “idols” are registered in different Southwestern Iberian Peninsula sites, being more common in funerary contexts [65,68,75]. Although not as common as the “idols” made in equid proximal phalanges, we have to mention the existence of several phalange idols made with red deer proximal phalanges at Perdigões (Figure 4), Santa Justa, and São Pedro [60,61]. Part of a wider phenomenon (see [77]), we should also mention the well-known depiction of deer or their antlers in different bell beaker pottery decorations, mostly in funerary but also non-funerary contexts from Portugal. These were found at Quinta do Anjo (also known as Casal do Pardo and Covas da Moura—Figure 4) [67,78], the tholos of Tituaria [79], Castro da Portucheira [80], Freiria [81], Perdigões [65], and Quinta do Estácio 14 [82], with another possible case at Leião [83].

## 4. Discussion

To approach human–deer relations during Late Prehistory, one must look at the different dimensions in which deer partook in these groups’ daily lives and how they or objects made with their body parts gained cultural agency or even personhood over time [84]. To disassociate the different spheres is to examine the parts. We are aware of the difficulties or even impossibility of understanding the different roles that deer played in the relational world of Late Prehistory. Several codifications could develop based on an animal’s ecology and ethology, and these characteristics could result in other relationships and ramifications [2,85], the archaeological visibility of which is very tenuous.

We intended to put into discussion the different zooarchaeological aspects of deer presence in the archaeological records of Central and Southern Portugal’s Late Prehistory, including “meaningful” and structured depositions, but combining this with the use of deer and deer body parts for technological, socio–cultural and ideological display. Previous studies have approached cervids in this region’s Late Prehistory but focus either on faunal data from a strictly economic perspective [34] or on other pieces of evidence, commonly rock art and other “artistic” expressions [9,66]. These, even if occasionally mentioning zooarchaeological information, do not consider the intricacies and variability of this data beyond the fact that deer were hunted.

### 4.1. On Deer in Artistic Expressions and Faunal Spectra

Cervids were of relevance for hunter–gatherers in Portugal. The “artistic” record from northern Portugal shows that red deer are the most represented animals in Late Azilian rock art engravings (45% [86]) and the Fariseu mobile art (61% [87]). In our area of interest, the Tagus Valley pre-schematic cycle, recently associated with the Mesolithic, is also dominated by cervids (58%) [5].

As for deer frequencies in the zooarchaeological records, we are dealing with different types of sites in different regions, with a varying proportion of sites/contexts excavated. Hence, we have a bias in the information, but within the scope of our approach, we do not consider it relevant enough to decisively hinder the discussion. Deer hunting, mostly red deer, was relevant for Mesolithic hunter–gatherers, especially towards the end of this period, as attested by different Late Mesolithic shell middens zooarchaeological records, even reaching values of 80% [36,88,89,90,91]. Davis and Detry [92] suggested a link between a smaller size of aurochs, red deer, and wild boar during the Mesolithic and their overexploitation. The larger size of these species during the Chalcolithic period could indicate a relaxation in hunting pressure. To what extent this could be applied to both Estremadura and Alentejo remains to be discussed when the amount of available data allows for that. However, this could have impacted human–deer relations as well.

The Tagus Valley schematic cycle (Neolithic, Chalcolithic, and Bronze Age) mainly comprises undetermined motifs, with cervids being the most common among identifiable animal figures [5]. Late Prehistoric cervid engravings have also been registered in Southern Portugal, in the Guadiana area. The authors suggest that the majority of these engravings date to the transition from the 4th to the 3rd millennium BCE [93]. Painted cervid figures are known, as is the case of Lapa dos Gaivões [94], but they are also found in the northern areas of Portugal, with cervids or deer hunting scenes depicted in megalithic dolmen orthostats [95,96,97,98,99]. Red deer associated with sun representations have been described in pottery and rock art motifs (e.g., [5,22]), and the possible association between the sun and polypod deer was recently discussed [15].

Within the Early Neolithic period, larger, more coastal settlements had small to no deer input, while slightly more inland sites at higher altitudes had a higher frequency. Sites located at lower altitudes and possibly related to a more stable occupation had a lower incidence of deer than the more sporadic ones from the Limestone Massif. This seems to be even clearer in the Middle Neolithic samples, which are dominated by caprines with a complementary input of deer, or vice-versa. Located on a different landscape unit and with its specificities due to being a ditched enclosure, Perdigões has a different faunal spectrum, but deer continue to appear in high quantities. The development of agro–pastoralism started to become clearer during the Late Neolithic, with a growth in the proportions of domesticated species. During this period, larger settlements from Estremadura lacked deer or had vestigial values, but some sites showed higher frequencies in Alentejo, which became apparent throughout the Chalcolithic.

During the 3rd millennium BCE, Estremadura, especially the sites located in its lower portion, showed vestigial or low deer frequencies. In contrast, the Alentejo records, especially those from larger sites, including ditched enclosures, showed higher deer frequencies. This could relate to the need to maintain larger populations, with domesticated animals being a more reliable food source. Alternatively, it could also relate to factors such as availability [34], sociocultural and ideological preferences, or the type of sites/contexts. At the same time, it is interesting that Vila Nova de São Pedro, where deer hunting shows an intermediate prevalence compared to all mentioned faunal spectra, is also where cervids were depicted in plaques [66].

Regarding availability, this is an issue that is difficult to discuss without paleo–vegetation records covering the periods and roughly spanning the different geographical areas under discussion. Because of this, we cannot discard the diachronic prevalence of deer species in the different regional records through time as also relating to availability. The different abundances of deer species, with an absence/low quantity of roe deer in comparison to red deer, could probably relate to this as well. Still, these issues do not invalidate the dynamics between subsistence and other practices since they can reinforce them.

Overall, deer were more prevalent in the Alentejo records during the Chalcolithic period. This is also a much richer area concerning other types of “meaningful” contexts, with social practices, in addition to more commonly accepted economic ones, being increasingly discussed. Possible feasting events are proof of this, with red deer body parts being included in larger wild animals’ food-sharing events or even being the most numerous species. No clear relation seems to exist so far between chronology within the Chalcolithic period and deer abundances for Estremadura. Still, some sites in the Alentejo show a higher prevalence of deer hunting towards the end of the 3rd millennium BCE, possibly related to the proposed breakdown in the trajectory of social complexity [54].

Although not circumscribed to the Iberian Peninsula, another aspect related to the Chalcolithic period, especially the end of the 3rd millennium BCE, is the Bell Beaker phenomenon. Bell beaker pottery decorated with cervids or, most notably, their antlers are found in several mentioned Portuguese sites but also in Spain (e.g., [77,100,101,102]), which is of interest to this research. This is not the appropriate place to discuss the contexts or findings and use of these pottery elements further, but the depictions follow the same standards seen in other Late Prehistoric artistic expressions (e.g., [5,9,66]).

### 4.2. On the Many Uses of Deer Body Parts

The use of deer teeth to adorn the body and decorate clothing or in connection with other artifacts is widespread (e.g., [103,104,105,106]. They have been considered to have a multifaceted meaning [103], with their reshaping being a transformation that integrates these remains (and their corresponding animals) into an artificial world of humans. In this “world” they remain “wild” but are also transformed into a different and domesticated form [103,104]. The mentioned cases for Early Neolithic pendants made from red deer teeth can be understood as similar behavior in agropastoral societies, where hunting, especially of red deer, seems to have maintained some relevance that would change mostly during the second half of the 4th millennium BCE, with the Late Neolithic and the Chalcolithic periods.

On this note, Chalcolithic “idols” are of interest (for recent synthesis on Portuguese data, see [107,108]). Focusing on phalange idols, due to the quality and quantity of information, we must mention the case of Perdigões [65]. Here, several phalange idols were recovered, with the prevalence of red deer phalange idols decreasing towards the end of the Chalcolithic period in favor of equid phalange idols. These are mostly shaped and rarely decorated, but different from what is seen elsewhere in Europe, where they can be found perforated (e.g., [106]). Cervids proliferated in the Alentejo zooarchaeological record during the 3rd millennium BCE and played a significant role in “meaningful” events. However, equids also started to gain relevance in the economic and non-economic spheres, the former in the Alentejo and the latter supra-regionally. The sole case of an ivory deer zoomorphic figurine from Perdigões is also noteworthy. Clay deer figurines are known in European Late Prehistory, being abundant in some cultures [106]. Ivory was used as a raw material to make other idols in Late Prehistoric Iberia. Despite being allochthonous, its rarity makes it relevant, if not symbolic, at least economically.

We are interested mostly in the questions surrounding antlers because several authors have already mentioned that the changing of antlers is related, for example, to the idea of “renovation” (e.g., [2]). Ethnographic and historical data show behaviors such as the burial of deer antlers and bones in “meaningful” places [3]. Deer antlers and other body parts recovered in Chalcolithic funerary contexts have been interpreted as depositions (e.g., [109,110,111]), but their relevance in funerary contexts was already evidenced with hunter–gatherers, together with artifacts made from deer antlers, bones, and teeth (e.g., [112,113]). Antlers can be collected after annual shedding, from a dead animal, or after hunting episodes, hence with different motifs and eventual meanings [114]. The lack of information on this specific topic, i.e., if the antlers were gathered or were from dead/killed animals, is common in archaeological data, thus hindering this type of discussion, but this would be worth pursuing in future research.

Based on ethnographic information, deer antlers (and skin) were used in deer hunting to facilitate the hunt or through the principle of “participation” [3]. Deer frontlets with antlers have been recovered in several Mesolithic sites, most notably Starr Carr, with Clark [1] relating them to hunting and other ritual practices. Conneller (p. 42, [115]) criticized this perspective, which was considered to be based on Western thought dichotomies (humans or animals, humans or things, stability or immutability of bodies). Based on anthropological accounts, animals can be understood as an assemblage of bodily effects, even part of broader assemblages of “affects”, with agency distributable through connections (p. 45, [115]). This changes the possible meaning of the creation and use of animal body parts, their agency, and “animalness”, which can be incorporated for various relational purposes. This includes changing bodily boundaries and perspectives, taking on the effects of the animal, and enabling engagement with the world in different ways.

Binford [116] stated that objects can have a primary functional context and he referred to objects related primarily to ideological functions as ideotechnic. Hodder [117] emphasized that material culture can have non-functional meanings. We agree with Valera (p. 240, [75]) that discriminating objects that assume symbolic, ideological principles are difficult, but their attributes, independently of context, may be related to the ideological sphere. Objects can have changeable status, and this category of “changeability” has also been associated with deer frontlets and antlers [3]. Animals have agency in social relationships and can be resources of a symbolic nature [118,119]. The deposition of animals and animal body parts can have a meaning beyond the economic sphere and even represent larger practices [114,120,121]. As mentioned, although rarer if compared to domesticated species, the deposition of near-to-complete or complete deer skeletons is registered in the regional Late Prehistory (for a Late Bronze Age example see [122]). The burial of ritually sacrificed deer has been reported elsewhere, with several complete skeletons and body parts being known in Europe (see synthesis in [123].

The seasonality of killing/deposition is also something that we cannot deepen in this discussion due to the general lack of information on published studies. Still, it seems that, in most cases, adults were included in these practices (cf. [122]). To the best of our knowledge, the deposition of complete deer skulls is absent, with only specific body parts considered structured depositions, mostly antlers. This hinders a more precise estimate of age at death or even sexing, beyond the presence of antlers, from red deer or roe deer, which are associated with male individuals. No more information has been published regarding the sexing of deer in the mentioned publications that could allow us to discuss how this influenced the decisions and inclusion of deer remains in the archaeological record. When looking at the described food-sharing events, only adult individuals are recorded [56], or mostly adults, but some long bones are still unfused without further description [57]. In addition, by being highly processed, these remains hinder more specific use of linear biometrics that could help to better characterize the specimens.

Variables such as sex, season of killing, selected deposition of body parts or entire skeletons, and the context (types of sites and deposits) in which these practices occur have been considered relevant (e.g., [123,124]). For example, Morris [124] indicates that chambered cairns could represent a boundary between the worlds of the dead and the living. The described inclusion of deer antlers in or near funerary contexts could have been related to this separation, in which the remains of deer and other animals could have functioned as liminal creatures in these liminal areas [124] or, as argued by [84], the personhood of animals and their modified or unmodified remains could relate to the identity and individuality of the living animal and their relationships with humans. Animal body parts, even if subject to modification, could be imbued with requests through communicative processes between humans and the remains before deposition. The personhood, individuality, and capabilities of these objects can be ritually acknowledged during the performative act of depositing them [84]. Deposition practices of animal body parts could even retain a sense of their “animalness” [115].

Even data more closely related to subsistence is of interest to this discussion. In Estremadura, where deer hunting was not as widespread, we find indicators of symbolism related to cervids. This raises questions about the possibility of deer hunting being associated with other practices or even becoming more performative as a consequence of its reduced relevance in subsistence. Russell [125] discussed the distinction between wild and domestic animals occurring in the Near Eastern Neolithic, impacting subsistence and ritual practices and resulting in different forms of personhood.

The majority of sites that could potentially be used to discuss these issues in Portugal have been excavated in the last three decades, accompanying a rise in contract-based archaeology that was not proportionally followed by a growth in zooarchaeological studies and context-specific discussion. Thus, a more descriptive approach is generally lacking. The existence of other “special “depositions (sensu [124]) has to be considered, and future works can help discuss human–deer relations based on more examples.

## 5. Conclusions

We aimed to discuss human–deer relations in Central and Southern Portugal’s Late Prehistory. Information from different regions and chronologies, funerary and non-funerary sites and contexts, and the use of deer remains for economic and other practices, including zooarchaeological and broader archaeological data, was mentioned and discussed at the supra-regional level. Animals and animal body parts can have agency, and that agency and its functions can change within the complexity of possible relationships, ontologies, and cosmogonies of Late Prehistoric groups. Variability is inherent to humans, and this impacts the relational meanings and narratives we construct with our surroundings, including animals, their parts, and representations. Based on the data presented and discussed, deer were a social element, with multiple and changing messages that are not accessible to us but that encompass several related non-funerary and funerary spheres and standardized practices, from structured depositions to food-sharing events and graphic representations.

## Figures and Tables

**Figure 1 animals-14-01424-f001:**
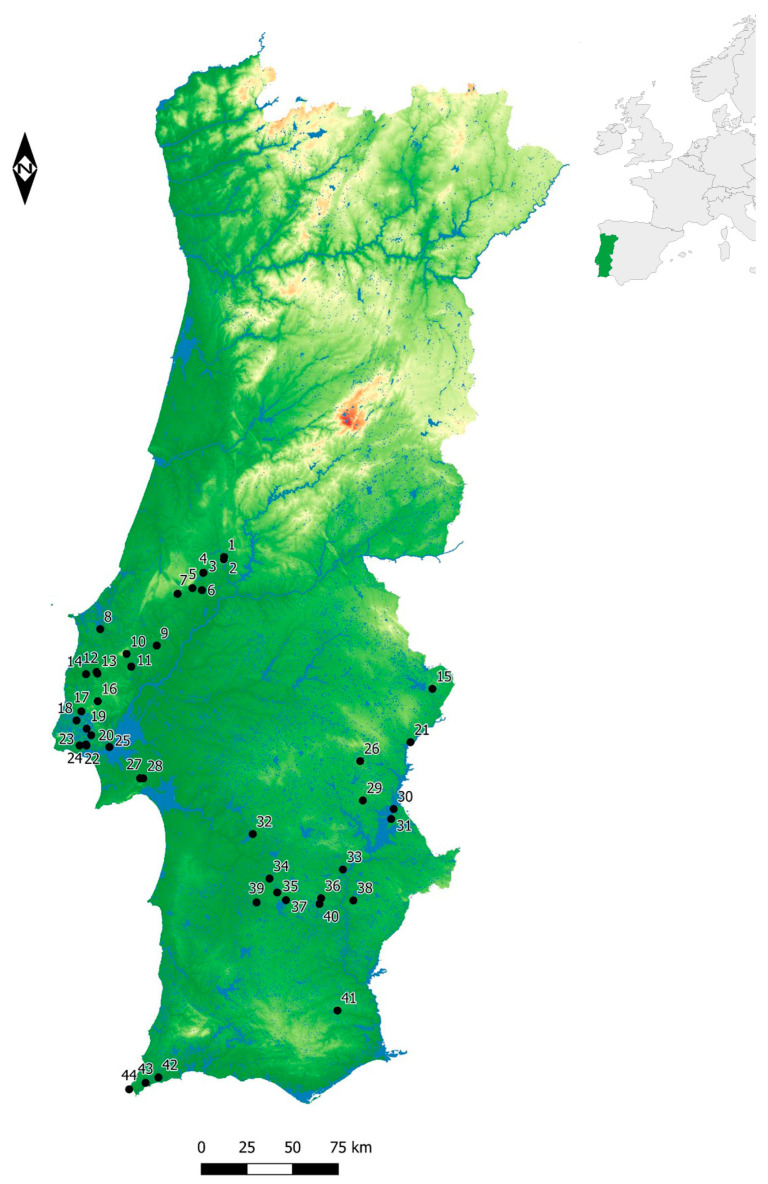
Map with the location of the sites mentioned in the text. The sites are as follows: 1—Cadaval, 2—Caldeirão, 3—Costa do Pereiro, 4—Pena d’Água, 5—Almonda (Galeria da Cisterna), 6—Nª Sª das Lapas, 7—Casais da Mureta, 8—Columbeira, 9—Vila Nova de São Pedro, 10—Bom Santo, 11—Ota, 12—Fórnea, 13—Portucheira, 14—Zambujal, 15—Santa Vitória, 16—Tituaria, 17—Penedo do Lexim, 18—Lameieas, 19—Belas, 20—Espargueira/Serra das Éguas, 21—Juromenha, 22—Leceia, 23—Freiria, 24—Carrascal, 25—Ecosta de Sant’Ana, 26—São Pedro, 27—Quinta do Anjo, 28—Chibanes, 29—Perdigões, 30—Mercador, 31—Moinho de Valadares, 32—Monte da Tumba, 33—Monte das Aldeias, 34—Porto Torrão, 35—Vale Frio 2, 36—Quinta do Estácio 14, 37—Monte do Outeirinho Novo, 38—Alto de Brinches 3, 39—Barranco do Xacafre, 40—Monte das Cabeceiras 2, 41—Santa Justa, 42—Vale Boi, 43—Padrão, 44—Rocha das Gaivotas.

**Figure 2 animals-14-01424-f002:**
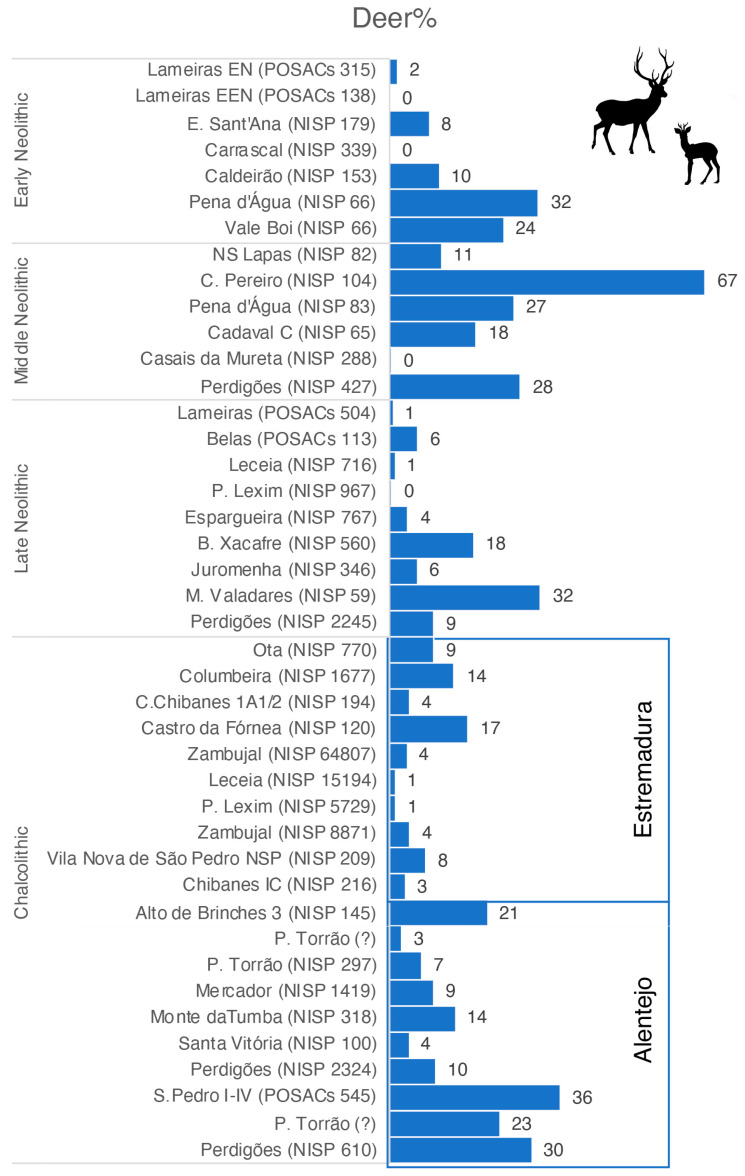
The ratio of cervids compared to *Equus*, *Bos*, *Sus*, and *Ovis*/*Capra* from different sites and phases, according to data from Table 1.

**Figure 4 animals-14-01424-f004:**
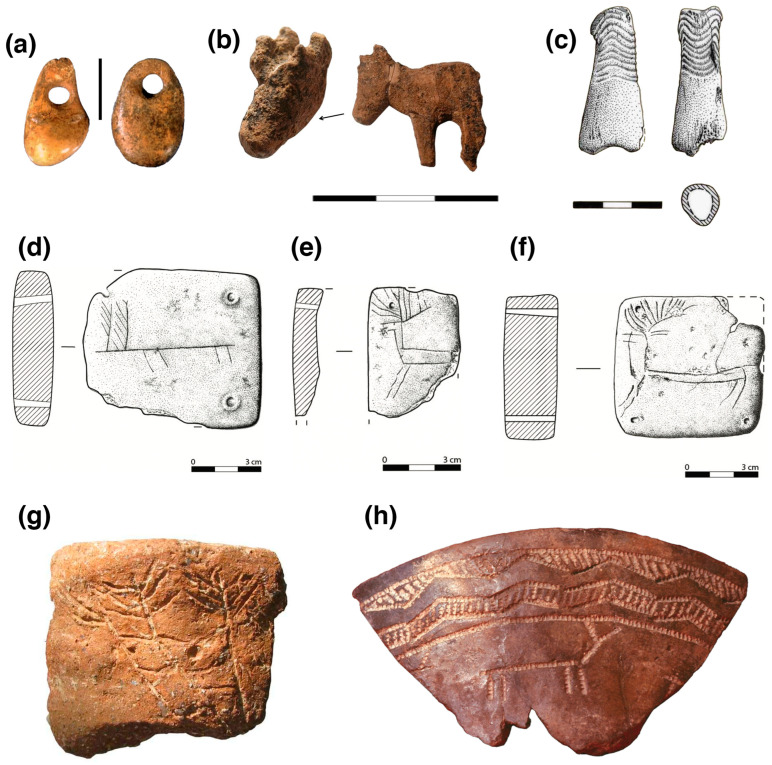
Some examples mentioned in the text are as follows: (**a**) Perforated red deer canine and bone bead imitation from Almonda (Galeria da Cisterna, adapted from [63]); (**b**) cervid figurine made in ivory from Perdigões tomb 2 (adapted from [61]); (**c**) drawing of a slightly shaped and decorated red deer proximal phalange from Perdigões tomb 1 (adapted from [65]). (**d**–**f**) Drawings by César Neves of a selection of Vila Nova de São Pedro plaques decorated with cervid motifs: (**d**) VNSP3000/MAC/236, (**e**) VNSP300/MAC/269, and (**f**) VNSP300/MAC/475 [66]; (**g**) pottery fragment with representation of cervids from Perdigões ditch 7 (no scale, adapted from [65]); (**h**) pottery fragment including decoration with cervids motifs from Quinta do Anjo (no scale, photo from E. Gameiro [67]).

**Table 1 animals-14-01424-t001:** Main non-funerary zooarchaeological assemblages with a macrofauna number of identified specimens (NISP) > 50 remain published for Central and Southern Portugal. NISPt = total NISP for *Equus*, *Bos*, *Sus*, *Cervus*/*Capreolus*, *Ovis*/*Capra*; NISPcerv = NISP for *Cervus*/*Capreolus.* EN = Early Neolithic, MN = Middle Neolithic, LN = Late Neolithic, CH = Chalcolithic. ^ Concerns assemblages chronologically framed within the end of the Chalcolithic or transition to the Early Bronze Age. * Assemblages where PoSACs are used instead of NISP; ** Estimate based on available PoSACs % (Parts of the Skeleton Always Counted). *** Data for the NISP from Porto Torrão are not given. The results correspond to % of an unknown number.

Site	Region	Phase	NISPt/NISPcerv	Reference
Lameiras *	Estremadura	EN	315/5	[16]
Lameiras *	Estremadura	EEN	138/0	[16]
Encosta de Sant’Ana	Estremadura	EN	179/15	[17]
Carrascal	Estremadura	EN	339/0	[18]
Caldeirão cave	Estremadura	EN	153/16	[19,20]
Pena d’Água rock-shelter	Estremadura	EN	66/21	[21,22,23]
Vale Boi	Algarve	EN	66/16	[24]
Nª Sª das Lapas cave	Estremadura	(EN)MN	82/9	[25]
Costa do Pereiro	Estremadura	MN	104/70	[26]
Pena d’Água rock-shelter	Estremadura	MN	83/22	[21,23,27]
Cadaval cave (layer C)	Estremadura	MN	65/12	[25]
Casais da Mureta	Estremadura	MN	288X/1	[28]
Perdigões	Alentejo	MN	427/119	[29]
Lameiras *	Estremadura	LN	504/3	[16]
Belas *	Estremadura	LN	113/5 **	[16]
Leceia	Estremadura	LN	716/7	[30]
Penedo do Lexim	Estremadura	LN	967/4	[31]
Espargueira/Serra das Éguas	Estremadura	LN(CH)	767/30	[32]
Barranco do Xacafre	Alentejo	LN	560/100	[33]
Juromenha 1	Alentejo	LN	346/20	[34,35]
Moinho de Valadares	Alentejo	LN	59/19	[36]
Perdigões	Alentejo	LN	2245/193	[28,37]
Ota	Estremadura	CH	770/72	[38]
Columbeira	Estremadura	CH	1677/231	[39]
Castro de Chibanes (1A1/2)	Estremadura	CH	194/8	[40]
Castro da Fórnea	Estremadura	CH	120/20	[41]
Castro do Zambujal	Estremadura	CH	64,807/2298	[42]
Leceia	Estremadura	CH	15,194/155	[30]
Penedo do Lexim	Estremadura	CH	5729/70	[43]
Castro do Zambujal	Estremadura	CH ^	8871/377	[42]
Vila Nova de São Pedro	Estremadura	CH ^	209/16	[44]
Castro de Chibanes (IC)	Estremadura	CH ^	216/7	[45]
Alto de Brinches 3	Alentejo	CH	145/30	[46]
Porto Torrão ***	Alentejo	CH	?/3%	[47]
Porto Torrão	Alentejo	CH	297/20	[48]
Mercador	Alentejo	CH	1419/132	[49]
Monte da Tumba	Alentejo	CH	318/44	[50]
Santa Vitória	Alentejo	CH	100/4	[51]
Perdigões	Alentejo	CH	2324/226	[52,53,54]
São Pedro I–IV *	Alentejo	CH ^	545/198	[55]
Porto Torrão ***	Alentejo	CH ^	?/23%	[47]
Perdigões	Alentejo	CH ^	610/184	[52,54,56]

## Data Availability

Data is contained within the article.

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
