# Peer review of "Human–Deer Relations during Late Prehistory: The Zooarchaeological Data from Central and Southern Portugal in Perspective"

_animals, 2024, doi:10.3390/ani14101424_

Round 1
Reviewer 1 Report
Comments and Suggestions for Authors
The manuscript “Human-deer relations during Late Prehistory: the zooarchaeological data from Central and Southern Portugal in perspective” discusses the interesting topic of symbolic meaning deer held in Neolithic Portugal. However, this manuscript is not entirely convincing due to a lack of sufficient clarity and data support.
Paragraph 3.2. “Deer in “meaningful” assemblages and structured depositions” lacks clear examples of deer symbolism. The arguments for the symbolic value of deer are not well exposed. The manuscript refers to previous publications without critical analyzing their claims or even without adequately presenting the material upon which the idea of a symbolic value of deer at the sites is based. For instances, phrases such as “food-sharing or feasting events including deer remains in very different frequencies were suggested” (lines 194-195), “a feasting event was proposed…” (line 212), “naturally fallen antlers, were considered offerings” (lines 223-224) lack sufficient elaboration. It would be beneficial if the authors could expand this paragraph with more data and detail their arguments supporting the proposal of a deer symbolism in these depositions.
Paragraph 3.3. “Deer and other materialities” is more interesting and convincing than the previous paragraph.
The manuscript could be improved by including in the Discussion comparisons with clear examples of deer symbolism in Neolithic sites from other parts of Europe (Conneller 2004, Larsson 2006, 2013, Osipowicz 2023, Pasaric 2023, Vitezovic 2024).
Suggested references
Conneller, C., 2004. Becoming deer. Archaeological Dialogues 11, 37–56.
Larsson L. 2006. Tooth for a tooth for a grave. Tooth ornaments from the graves at the cemetery of Zvejnieki. In Larsson & Zagorska (eds) 2006, 253–87 Larsson, L. & Zagorska, I. (eds). 2006. Back to the Origin: new research in the Mesolithic–Neolithic Zvejnieki cemetery and environment, northern Latvia. Lund: Acta Archaeologica Lundensia Series in 8° 52.
Larsson, L., 2013. Tooth-beads, antlers, nuts and fishes: examples of social bioarchaeology. Archaeological Dialogues 20, 148–52.
Osipowicz, G., Orłowska, J., Zagorska, I., 2023. Towards understanding the influence of Neolithisation for communities using the Zvejnieki cemetery, Latvia: A technological and functional analysis of the osseous artefacts discovered in the Late Mesolithic burial no 57 and Neolithic burial no 164. Quaternary International 665–666, 65-81, https://doi.org/10.1016/j.quaint.2022.11.007 .
PASARIĆ M., 2023. ‘Come and Give my Child Wit’. Animal Remains, Artefacts, and Humans in Mesolithic and Neolithic Hunter-gatherer Graves of Northern Europe. Proceedings of the Prehistoric Society 89, 207-224. doi:10.1017/ppr.2023.1
Vitezovic, S., 2024. Taking the Deer by the Antlers: Deer in Material Culture in the Balkan Neolithic. Arts 13: 64. https://doi.org/10.3390/arts13020064
2.12.0.0 2.12.0.0 Comments on the Quality of English LanguageThe English of the manuscript is sometimes confusing. The Discussion includes the following sentence: “We are aware of the difficulties or even impossibility of understanding the different papers that deer had in the relational world of Late Prehistory.” (line 296-298). However, the meaning of “papers” in this context is unclear.
2.12.0.0 2.12.0.0Author Response
The authors would like to thank in advance to the reviewer for his comments and suggestions that helped considerably improve the paper submitted. The reviewer's critics are addressed below point-by-point.
The manuscript “Human-deer relations during Late Prehistory: the zooarchaeological data from Central and Southern Portugal in perspective” discusses the interesting topic of symbolic meaning deer held in Neolithic Portugal. However, this manuscript is not entirely convincing due to a lack of sufficient clarity and data support.
- More information and needed clarifications were added, as well as a deeper discussion. Hopefully, the reviewer will consider this improved version as having more clarity and data support.
Paragraph 3.2. “Deer in “meaningful” assemblages and structured depositions” lacks clear examples of deer symbolism. The arguments for the symbolic value of deer are not well exposed. The manuscript refers to previous publications without critical analyzing their claims or even without adequately presenting the material upon which the idea of a symbolic value of deer at the sites is based. For instances, phrases such as “food-sharing or feasting events including deer remains in very different frequencies were suggested” (lines 194-195), “a feasting event was proposed…” (line 212), “naturally fallen antlers, were considered offerings” (lines 223-224) lack sufficient elaboration. It would be beneficial if the authors could expand this paragraph with more data and detail their arguments supporting the proposal of a deer symbolism in these depositions.
- Several additions were made regarding these issues in the results and discussion sections.
Paragraph 3.3. “Deer and other materialities” is more interesting and convincing than the previous paragraph.
The manuscript could be improved by including in the Discussion comparisons with clear examples of deer symbolism in Neolithic sites from other parts of Europe (Conneller 2004, Larsson 2006, 2013, Osipowicz 2023, Pasaric 2023, Vitezovic 2024).
- A large set of bibliography was added, including publications suggested by the reviewer and improving the discussion section.
Reviewer 2 Report
Comments and Suggestions for Authors
The article succinctly and appropriately describes the evolution of human-deer relations throughout the late prehistoric periods in the Iberian Peninsula; both as a source of food and as a motif in the spiritual life, culture and ancient art, while cross-referencing valuable information from research sites in central and southern Portugal. The synthesis of the available data makes it clear that human-deer relations have changed over time and space. The utilization of the deer in its entirety or alternatively the utilization of certain parts of their body as a resource of a symbolic nature also changed, and was included in public events of sharing food, burial offerings, as well as in artistic representations. The fascinating change in the attitude of the ancient human society to the deer receives important publicity in this work.
In summary, this is a good and publishable work, yet several elements of the work should be improved:
1- The introduction is well written, yet it would be helpful to add a picture of the major deer species discussed. Also, the authors should expand on the ecology of the deer, which may be a reason for their frequency in some of the studied sites.
2- The method of collecting the faunal remains at the various sites - whether from sifting of the sediment or only from manual collection must specified by the authors.
3- The results are well detailed.
4- The discussion is appropriate, but it is necessary to cross-reference information between the results and the deer ecology and the methods of collecting the information at the research sites.
Author Response
The authors would like to thank in advance to the reviewer for his comments and suggestions that helped considerably improve the paper submitted. The reviewer's critics are addressed below point-by-point.
The article succinctly and appropriately describes the evolution of human-deer relations throughout the late prehistoric periods in the Iberian Peninsula; both as a source of food and as a motif in the spiritual life, culture and ancient art, while cross-referencing valuable information from research sites in central and southern Portugal. The synthesis of the available data makes it clear that human-deer relations have changed over time and space. The utilization of the deer in its entirety or alternatively the utilization of certain parts of their body as a resource of a symbolic nature also changed, and was included in public events of sharing food, burial offerings, as well as in artistic representations. The fascinating change in the attitude of the ancient human society to the deer receives important publicity in this work.
In summary, this is a good and publishable work, yet several elements of the work should be improved:
1- The introduction is well written, yet it would be helpful to add a picture of the major deer species discussed. Also, the authors should expand on the ecology of the deer, which may be a reason for their frequency in some of the studied sites.
- A short paragraph on the ecology of the deer species discussed was added to the introduction section. Since only two species exist, red deer and roe deer, and the latter is almost non-existent in the faunal records, the authors see no need to add a picture of only these two species. If the reviewer thinks this is imperative, we will consider this situation.
2- The method of collecting the faunal remains at the various sites - whether from sifting of the sediment or only from manual collection must specified by the authors.
- Information regarding these issues was added to the Materials and Methods.
3- The results are well detailed.
- Many thanks for your comment.
4- The discussion is appropriate, but it is necessary to cross-reference information between the results and the deer ecology and the methods of collecting the information at the research sites.
- The discussion was subject to several additions and clarifications, among these, also the issues indicated by the reviewer were included.
Reviewer 3 Report
Comments and Suggestions for Authors
I am not qualified to assess the quality of English, but some words sound inaccurate to me ("phalange", "made on bone..."); is “meaningful” assemblages the best term?
The strength of the manuscript is that it summarizes all available information of various types on deer in a defined region and period.
The weakness of the text lies in the fact that it is more or less descriptive text and that, based on the combination of diverse sources and data on findings, the manuscript does not bring a new vision that would advance our idea of the role of the deer.
However, it was not the stated aim of the authors and I rate the given summary and combination of findings as very useful for many readers from various research disciplines. And I recommend it for publishing with just a few suggestions:
- Since the words "bone" and "osteology" are not mentioned in the Title and Abstract, it could be useful to involve one of them in Keywords.
- Please, explain the acronym PoSACs in Methods or somewhere.
- It could be useful to formulate a short and clear conclusion: Is there any new observation or answer by the authors that can be highlighted in the Conclusion?
- Some aspects were not exploited fully by the authors such as a variety of utilization of deer in ethnographical “analogies“ or deer sex, individual age, and seasonality of killing/deposition, which can be easily readable from bones and could play a role in interpretation (see, e.g., https://link.springer.com/article/10.1007/s12520-016-0344-x ). Is there some meaning in terms of killed and non-killed (gathered antlers) animals, imports (ivory), etc.
- I suggest to make the picture in Fig. 4 a bit bigger (wider). Now details can be difficult to see.
Author Response
The authors would like to thank in advance to the reviewer for his comments and suggestions that helped considerably improve the paper submitted. The reviewer's critics are addressed below point-by-point.
I am not qualified to assess the quality of English, but some words sound inaccurate to me ("phalange", "made on bone..."); is “meaningful” assemblages the best term?
- The English language was revised throughout the paper. Phalange is correct, and we believe that "meaningful" is a terminology that corresponds to our interpretation of these assemblages in comparison to terms used by other colleagues as "special" assemblages (Morris 2005).
The strength of the manuscript is that it summarizes all available information of various types on deer in a defined region and period.
The weakness of the text lies in the fact that it is more or less descriptive text and that, based on the combination of diverse sources and data on findings, the manuscript does not bring a new vision that would advance our idea of the role of the deer.
- We believe the modifications made in the manuscript resulted in a more interesting and relevant discussion that hopefully the reviewer will think are a good addition to the existing bibliography.
However, it was not the stated aim of the authors and I rate the given summary and combination of findings as very useful for many readers from various research disciplines. And I recommend it for publishing with just a few suggestions:
Since the words "bone" and "osteology" are not mentioned in the Title and Abstract, it could be useful to involve one of them in Keywords.
- Osteology was added to keywords.
Please, explain the acronym PoSACs in Methods or somewhere.
- Added to the Table caption.
It could be useful to formulate a short and clear conclusion: Is there any new observation or answer by the authors that can be highlighted in the Conclusion?
- A conclusion paragraph was added.
Some aspects were not exploited fully by the authors such as a variety of utilization of deer in ethnographical “analogies“ or deer sex, individual age, and seasonality of killing/deposition, which can be easily readable from bones and could play a role in interpretation (see, e.g., https://link.springer.com/article/10.1007/s12520-016-0344-x ). Is there some meaning in terms of killed and non-killed (gathered antlers) animals, imports (ivory), etc.
- More data and a considerably improved discussion are presented in this new version where these and other issues are included.
I suggest to make the picture in Fig. 4 a bit bigger (wider). Now details can be difficult to see.
- Fig. 4 size was changed accordingly.
Round 2
Reviewer 1 Report
Comments and Suggestions for Authors
The authors have taken into account my suggestions. Their revised manuscript is now ready for publication.
2.12.0.0 2.12.0.0 2.12.0.0